# Forced Overexpression of Signal Transducer and Activator of Transcription 3 (STAT3) Activates Yes-Associated Protein (YAP) Expression and Increases the Invasion and Proliferation Abilities of Small Cell Lung Cancer (SCLC) Cells

**DOI:** 10.3390/biomedicines10071704

**Published:** 2022-07-14

**Authors:** Ping-Chih Hsu, Jhy-Ming Li, Cheng-Ta Yang

**Affiliations:** 1Department of Thoracic Medicine, Chang Gung Memorial Hospital at Linkou, Taoyuan City 33305, Taiwan; 8902049@adm.cgmh.org.tw; 2Department of Medicine, College of Medicine, Chang Gung University, Taoyuan City 33302, Taiwan; 3Division of Colon and Rectal Surgery, Department of Surgery, Chang Gung Memorial Hospital at Chiayi, Chiayi County 61363, Taiwan; jml@mail.ncyu.edu.tw; 4Department of Animal Science, National Chiayi University, Chiayi City 60004, Taiwan; 5Department of Internal Medicine, Taoyuan Chang Gung Memorial Hospital, Taoyuan City 33378, Taiwan; 6Department of Respiratory Therapy, College of Medicine, Chang Gung University, Taoyuan City 33302, Taiwan

**Keywords:** signal transducer and activator of transcription 3 (STAT3), yes-associated protein (YAP), Hippo signaling, small cell lung cancer, epithelial–mesenchymal transition (EMT), proliferation

## Abstract

Background: We sought to investigate the interaction between signal transducer and activator of transcription 3 (STAT3) and the Yes-associated protein (YAP) signaling pathway in human small cell lung cancer (SCLC) cells. Methods: The STAT3-overexpressing SCLC cell lines H146 and H446 were established by plasmid DNA transfection for in vitro and in vivo experiments. Results: Overexpression of STAT3 increased YAP protein expression in H146 and H446 cells. STAT3 overexpression significantly increased YAP mRNA expression and the mRNA expression of the YAP signaling downstream genes CTGF and CYR61 in H146 and H446 cells (*p* < 0.05). We showed that STAT3 overexpression promoted EMT (epithelial–mesenchymal transition) with increased matrix metalloproteinase (MMP)-2 and MMP9 expression. Transwell assays showed that STAT3 overexpression increased the invasion ability of H146 and H446 cells. In addition, STAT3-overexpressing H146 cells grew significantly more rapidly than control H146 cells in the xenograft mouse model (*p* < 0.05). Immunohistochemistry (IHC) staining and Western blotting (WB) showed that STAT3-overexpressing H146 tumors had increased p-STAT3 and YAP staining and protein expression compared with control tumors. Increased EMT was also observed in STAT3-overexpressed xenograft tumors. Conclusions: The results of our study suggest that the overexpression of STAT3 promotes SCLC EMT, invasion, and proliferation through the activation of the YAP signaling pathway.

## 1. Introduction

Small cell lung cancer (SCLC) accounts for approximately 10–15% of the histological types of primary lung cancers [1,2]. SCLC exhibits extremely aggressive behavior with rapid growth and early spread to distant sites, and thus extensive-stage (metastatic) disease appears in most patients at the initial diagnosis [3,4]. The prognosis of extensive-stage SCLC is extremely poor, with a one-year survival rate of 27% and a two-year survival rate of 8% [1]. The therapeutic options for extensive-stage SCLC are very limited, and conventional chemotherapy is still the main treatment [3,4]. Although anti-programmed death-ligand 1 (PD-L1) immune checkpoint inhibitors (ICIs) (atezolizumab and durvalumab) in addition to chemotherapy have successfully improved the survival of extensive-stage SCLC in pivotal clinical trials (Impower133 and CASPIAN trials), the overall survival was significantly prolonged by only 2 months when compared with the control group patients [5,6]. To date, no effective targeted therapy has been developed and used for the treatment of extensive-stage SCLC [7]. An understanding of the molecular and biological mechanisms of tumorigenesis and cancer progression in SCLC has emerged.

Signal transducer and activator of transcription 3 (STAT3) is an important mediator of the interleukin-6 (IL-6)/Janus tyrosine kinase (JAK)/STAT3 signaling pathway and has been shown to promote tumorigenesis in various cancers, including lung cancer, lymphoma, pancreatic cancer and hepatocellular carcinoma [8,9]. Previous studies have shown that STAT3 is involved in tumor proliferation and drug resistance in small cell lung cancer. One previous study demonstrated that EPHA3 was associated with drug resistance and prognosis in SCLC patients, and inhibition of EPHA3 decreased STAT3 protein expression in SCLC cells [10]. Another study conducted by Zhu et al. showed that the knockdown of KCNQ1OT1 inactivated the JAK2/STAT3 signaling pathway and suppressed the migration, invasion and tumor growth abilities of SCLC cells [11].

Yes-associated protein (YAP) is one of the main mediators of the Salvador–Warts–Hippo (or simply Hippo) signaling pathway. The hyperactivation of YAP is frequently found in various cancers and has been suggested to be involved in cancer initiation and progression [12,13]. In human non-small cell lung cancer, YAP has been identified as an oncogenic protein that promotes drug resistance, tumorigenesis and metastasis in several previous studies [13,14,15,16]. YAP is negatively regulated by Hippo kinases, such as Merlin, large tumor suppressor homolog 1 (LATS1), LATS2, and mammalian sterile 20-like kinase 1 (MST1). In the usual manner, the Hippo kinases sequester and degrade YAP in the cytoplasm. Conversely, when the Hippo kinase is deregulated, the increasing YAP translocates from the cytoplasm into the nucleus and binds to transcriptional enhancer factors (TEAD). The binding of YAP and TEAD forms a complex, enhancing the downstream gene transcription to promote cell proliferation, EMT, invasion and drug resistance [13,14,15,16]. In a previous study, YAP expression was also identified in certain phenotypes of human SCLC [17]. Another recent study reported that YAP may be associated with radiation resistance in SCLC [18].

Previous studies reported that STAT3 interacted with YAP signaling in head and neck squamous cell carcinoma (HNSCC) and colorectal cancer (CRC). Li et al. showed that STAT3 activated YAP to promote tumor progression through suppressing the Hippo kinase LATS1 [19]. Another study conducted by Shen et al. concluded that the interaction of the STAT3–YAP axis plays a crucial role in angiogenesis in CRC cells [20]. Whether STAT3 crosstalks with the YAP signaling pathway in SCLC is unclear. In this study, we sought to explore the interaction between STAT3 and YAP, and the impacts of the overexpression of STAT3 on YAP signaling downstream genes, EMT, invasion, and proliferation in human SCLC cells.

## 2. Materials and Methods

### 2.1. Cell Culture

This study was performed mainly using SCLC cell lines. Most SCLC cell lines appear with a floating growth pattern (H146, H446 and H720), and H209 is a rare SCLC cell line with an adherent growth pattern. We selected these cell lines to verify whether there is different STAT3 and YAP expressions in SCLC with different growth patterns. The cell lines used in this study, including H146 and H209, were purchased from the Bioresource Collection and Research Center (BCRC, Hsinchu, Taiwan), and H446 and H720 cells were purchased from the American Type Culture Collection (ATCC). H146 cells were maintained in RPMI 1640 medium with 10 mM HEPES and 1.0 mM sodium pyruvate and supplemented with 10% fetal bovine serum (Life Technologies Corporation). H720 cells were cultured in DMEM/F-12 medium containing 2.5 mM L-glutamine, 15 mM HEPES, 0.5 mM sodium pyruvate and 1200 mg/L sodium bicarbonate, and the following components were added to the base medium: 0.005 mg/mL insulin, 0.01 mg/mL transferrin, 30 nM sodium selenite, 10 nM hydrocortisone, 10 nM beta-estradiol (Sigma–Aldrich, MO, USA), 2 mM l-glutamine and 5% fetal bovine serum. H209 and H446 cells were cultured in RPMI medium supplemented with 10% fetal bovine serum. All culture media used were supplemented with 1% penicillin/streptomycin and incubated at 37 °C in 5% CO_2_ and saturated humidity.

### 2.2. Establishment of SCLC Cell Lines H146 and H446 with Overexpression of STAT3, and STAT3 Small Interfering RNA (siRNA) Transfection

We established SCLC cell lines with stable STAT3 overexpression to investigate the impacts on YAP signaling, EMT, invasion ability and proliferation. H146 and H446 cells with stable STAT3 overexpression were established by using lentiviral packaging and infection. The supernatant was collected 48 h after cotransfection of the overexpression-STAT3 vector (addgene), and each packaging vector (pCMVdeltaR8.91 and pMD. G; National RNAi Core Facility Platform) into HEK 293T cells was centrifuged at 6000× *g* for 15 min. The pellet was dissolved in Dulbecco’s modified Eagle’s medium (Sigma) and stored at −80 °C. Following overnight culture of H146 cells at 1 × 10^5^ cells per well of a 12-well culture plate, the cells were infected with the supernatant and cultured overnight. After washing out the virus several times with PBS, H146 and H446 cells with STAT3 overexpression were selected by puromycin, and stemness expression was detected. STAT3 siRNA transfection was used to knock down STAT3 in H209 cells. STAT3 siRNA#1 and siRNA#2 were purchased from Life Technologies (Grand Island, NY, USA), and targeting sequences were GGCUGGACAAUAUCAUUG and UCAAUGAUAUUGUCCAGC. The STAT3 siRNAs were transfected using Lipofectamine RNAiMAX (Invitrogen, Carlsbad, CA, USA), and transfected cells were harvested after transfection for 48 h.

### 2.3. Animal Studies

An animal experiment with xenograft establishment was performed to verify the proliferation and EMT in STAT3-overexpressing H146 cells in vivo. All the in vivo experiments strictly followed the institutional guidelines of Chang Gung Memorial Hospital (Institutional Animal Care and Use Committee approval number: 2018032804). Athymic nude (CrTac: NCr-Foxn1nu) female mice 6–8 weeks of age were purchased from the National Laboratory Animal Center (Taipei, Taiwan). A xenograft murine model was established by injecting 5,000,000 parental H146 and STAT3-overexpressing H146 cells suspended in 50 µL of PBS mixed with an equal volume of Matrigel^®^ (Corning Inc., Oneonta, NY, USA) subcutaneously into the flanks of the mice. The tumor volumes were measured every 3 days after implantation in the mice. The tumor size was not allowed to exceed 2000 mm^3^, and mice were euthanized when the tumor size approached 2000 mm^3^ or when mice appeared extremely sick. After 21 days of the mouse experiment, the mice were sacrificed, and tumors were harvested for pathological analysis.

### 2.4. Western Blot Analysis

Western blotting (WB) was performed to explore the YAP and EMT marker protein expression changes in H146 and H446 cells after STAT3 overexpression. Total cell lysates from the SCLC cell lines were extracted with lysis buffer (M-PER Mammalian Protein Extraction Reagent; Thermo Fisher Scientific Inc., Rockford, IL, USA). The proteins were run on 4–20% gradient SDS–polyacrylamide gels (Bio–Rad Laboratories, Inc., Hercules, CA, USA) and then transferred to Immobilon-Pnitrocellulose membranes (Millipore, Bellerica, MA, USA). The membranes were probed with primary antibodies purchased from Cell Signaling Technology Inc., Danvers, MA, USA, including anti-stat3 (#30835), anti-p-STAT3 (#52075), anti-YAP (#8418), anti-LATS (#3477), 1anti-E-cadherin (#3195), anti-N-cadherin (#13116), ant-vimentin (#5741), and ant-β-actin (#5057). Horseradish peroxidase-conjugated goat anti-rabbit antibody was used as the secondary antibody (Santa Cruz Biotechnology). Chemiluminescence detection was carried out using ECL Plus (GE Healthcare, Piscataway, NJ, USA) and executed according to the manufacturer’s instructions.

### 2.5. RNA Isolation, cDNA Synthesis and Quantitative Real-Time Polymerase Chain Reaction (qRT–PCR)

qRT–PCR was performed to determine the YAP and downstream CTGF and CYR61 mRNA expression changes in H146 and H446 cells with STAT3 overexpression. The QIAamp RNeasy Mini kit (Qiagen, Valencia, CA, USA) was used for RNA extraction from SCLC cells. The cDNA used as the template for qRT–PCR was transcribed from RNA using iScript-cDNA Synthesis Kits (Bio-Rad, Hercules, CA, USA) according to the manufacturer’s protocol. An Applied Biosystems 7000 sequence detection system (Applied Biosystems, Foster City, CA, USA) was used for qRT–PCR detection. Expression levels of the YAP, CTGF, and CYR61 genes and GAPDH as an internal control were detected using commercial primer and probe sequences purchased from TaqMan Thermo Fisher (Thermo Fisher Scientific, Rockford, IL, USA) and analyzed using Relative Quantification Software (Applied Biosystems, Foster City, CA, USA). Quantification Software (Thermo Fisher Scientific, Waltham, MA, USA) was used for final mRNA expression analysis. The TaqMan assay IDs of primer sequences for qRT–PCR were GAPDH (Hs02786624_g1), YAP (Hs00902712_g1), CTGF (Hs01026927_g1), and CYR61 (Hs00155479_m1).

### 2.6. Assessment of Gelatinases Matrix Metalloproteinase-2 (MMP2) and Matrix Metalloproteinase-9 (MMP9) by Gelatin Zymography

Gelatin zymography was performed to verify the MMP2 and MMP9 changes in H146 and H446 cells after STAT3 overexpression. Control H146 and control H446 cells and STAT3-overexpressing H146 and STAT3-overexpressing H446 cells were incubated in serum-free media for 24 h and 48 h, respectively. The cell-conditioned media were then collected and concentrated 10 times using Amicon^®^ Ultra-4 centrifugal filters (Merck Millipore). The concentrated cell-conditioned media samples were loaded on SDS-polyacrylamide gels with 1 mg/mL gelatin and run at 150 voltage until band separation was achieved for analysis. The gels were stained with Coomassie Brilliant Blue G-250 (Sigma-Aldrich), and MMP2 and MMP9 activities were detected as transparent bands present on the blue background.

### 2.7. Transwell Invasion Assay

A Transwell assay was performed to investigate the invasion ability of H146 and H446 cells after STAT3 overexpression. A 6-well plate Transwell system (Corning Incorporated, New York, NY, USA) was used for the Transwell invasion assay, and the Transwell inserts were coated with 300 μL of diluted Matrigel (Matrigel/nonserum medium = 1:5, total volume 300 μL). Control H146, control H446, STAT3-overexpressing H146 and STAT3-overexpressing H446 cells were trypsinized and resuspended in serum-free medium, and the cells were seeded in the upper chamber of the Transwell. The lower chamber was infused with 2 mL of complete growth medium (10% FBS). The gel and cells in the upper chamber of the Transwell were wiped after incubation at 37 °C for 48 h. The membrane was stained with crystal violet for 10 min after methanol fixation. Phase-contrast images were captured using a Primo Vert microscope (ZEISS, Gottingen, Germany).

### 2.8. Immunohistochemistry (IHC) Staining of p-STAT3 and YAP

IHC staining was performed after the animal experiment to see the late effect of STAT3 overexpression in xenograft tumors. After the mice were euthanized, the tumor tissues of xenografts were harvested and fixed in 10% paraformaldehyde for 24 h. The xenograft tumors were stored in 70% ethanol and then embedded in paraffin. Rabbit anti-p-STAT3 antibody (Cell Signaling, #9145; 1:500) and rabbit anti-YAP antibody (Cell Signaling, #4912; 1:500) were used for immunohistochemistry (IHC) staining. The xenograft tumor tissue slides underwent deparaffinization and rehydration. The slides were immersed in 10 mM sodium citrate buffer (pH = 6.0) and boiled for 10 min on a hot plate. The slides were then allowed to cool for 20 min. After cooling, the slides were incubated in 3% hydrogen peroxide for 10 min and then washed with PBS every 5 min three times. After washing with PBS, the slides were incubated with 10% normal goat serum in PBS for 30 min and then washed with PBS again. The slides were immunostained with primary antibodies (YAP and STAT3) and incubated at 4 °C overnight. The slides were incubated with biotin-labeled secondary antibodies and streptavidin–peroxidase (1:30) for 20 min. The slides were stained for 5 min with 0.05% 3,3’-diaminobenzidine tetrahydrochloride freshly prepared in 0.05 M Tris–HCl buffer (pH = 7.6) containing 0.024% hydrogen peroxide, dehydrated and mounted in Diatex. All images were captured using a Primo Vert microscope (ZEISS, Gottingen, Germany).

### 2.9. Statistical Analysis

Statistical analysis was performed to determine the changes in mRNA expression and the quantitative analysis of the Transwell invasion assay. We used GraphPad Prism (Version 5.0; GraphPad Software, San Diego, CA, USA) to perform statistical analysis of the experimental data. Data are shown as the mean ± standard deviation (SD) from three independent experiments. Student’s t-test was used to analyze the differences between two groups. All *p* values were 2-sided and defined as statistically significant if *p* was less than 0.05 (* *p* < 0.05, ** *p* < 0.01, *** *p* < 0.001).

## 3. Results

### 3.1. STAT3 and YAP Expression in Small Cell Lung Cancer Lines

WB showed that H209 cells had increased p-STAT3 protein expression compared to the other three cell lines (H146, H446 and H720) (Figure 1A). In four SCLC cell lines (H209, H146, H446 and H720), YAP protein expression obviously appeared in H209 cells (Figure 1A). Quantitative analysis of the protein expression level showed that H209 cells had concurrently increased p-STAT3 and YAP protein expression compared with the other three SCLC cell lines (H146, H446 and H720) (Figure 1B,C). The morphologies of the SCLC cell lines are shown in Figure 1D. H209 cells with increased p-STAT3 and YAP protein expression exhibited an adherent growth pattern, and SCLC cell lines H146, H446, and H720 with low p-STAT3 and YAP protein expression exhibited a floating growth pattern (Figure 1D).

### 3.2. Forced Overexpression of STAT3 in H146 and H446 Cells

WB was used to verify the effect of STAT3 overexpression in H146 and H446 cells by STAT3 vector transfection. WB showed that STAT3-overexpressing H146 and H446 cells had increased STAT3, p-STAT3 and YAP protein expression compared with control H146 and H446 cells (Figure 2A–E). The morphologies of STAT3-overexpressing H146 and H446 cells transformed from floating to adherent growth patterns (Figure 2F,G). The mRNA expression levels of YAP and the YAP signaling downstream genes CTGF and CYR61 were assayed by qRT–PCR. YAP, CTGF, and CYR61 mRNA expression was significantly higher in STAT3-overexpressing H146 and H446 cells than in control H146 and H446 cells (*p* < 0.05) (Figure 2F,G). The results indicate that forced overexpression of STAT3 in H146 and H446 cells activated YAP protein expression and transformed the cell morphology. In addition, the overexpression of STAT3 increased the mRNA expression of YAP and the YAP signaling downstream genes CTGF and CYR61 in H146 and H446 cells. To verify the mechanism of how STAT3 increased YAP expression, the Hippo kinase LATS1 was assayed by WB. WB showed that LATS1 protein expression decreased in STAT3-overexpressing H146 and H446 cells compared with control H146 and H446 cells (Appendix A). The changes in YAP mRNA and YAP signaling downstream genes CTGF and CYR61 in H209 cells after STAT3 knockdown by siRNAs were also assayed by qRT–PCR. STAT3 knockdown by siRNAs did not decrease the mRNA level of YAP, CTGF, and CYR61 expressions in H209 cells (Appendix A). These results indicate that the forced overexpression of STAT3 activates the YAP protein by suppressing the Hippo kinase LATS1 and that the CTGF and CYR61 genes were mainly regulated by activating YAP, not STAT3.

### 3.3. Forced Overexpression of STAT3 Increased EMT Markers in H146 and H446 Cells

The effect of forced STAT3 overexpression on epithelial–mesenchymal transition (EMT) was assayed by WB and gel zymography. WB showed that STAT3-overexpressing H146 and H446 cells had decreased E-cadherin (epithelial marker) protein expression compared with control H146 and H446 cells. STAT3-overexpressing H146 and H446 cells had increased N-cadherin and vimentin (mesenchymal marker) protein expression compared with control H146 and H446 cells (Figure 3A). In gel zymography, STAT3-overexpressing H146 and H446 cells had increased matrix metalloproteinase-2 (MMP2) and matrix metalloproteinase-9 (MMP9) expression compared with control H146 and H446 cells (Figure 3B,C). The results suggest that the forced overexpression of STAT3 promoted EMT in SCLC H146 and H446 cells.

### 3.4. Forced Overexpression of STAT3 Increased the Invasion Ability of H146 and H446 Cells

Transwell assays were performed to investigate whether forced overexpression of STAT3 increased the invasion ability of H146 and H446 cells. Transwell assays showed that the number of cells invading the lower side of membranes obviously increased in STAT3-overexpressing H146 and H446 cells compared with control H146 and H446 cells by crystal violet staining (Figure 4A,C). In quantitative analysis of the Transwell invasion assay, the number of cells invading the lower side of membranes significantly increased in STAT3-overexpressing H146 and H446 cells (*p* < 0.05) (Figure 4B,D). Our results verify that the forced overexpression of STAT3 increased the invasion ability of H146 and H446 cells.

### 3.5. Forced Overexpression of STAT3 Increased the Cell Proliferation Ability of H146 Cells in a Mouse Model

To investigate whether the forced overexpression of STAT3 promotes tumor growth in vivo, we implanted STAT3-overexpressing and control H146 cells in the flanks of nude mice. STAT3-overexpressing H146 xenografts grew significantly more rapidly than control H146 cells in a xenograft mouse model (Figure 5A,B).

Xenograft tumors were collected from the control and STAT3 overexpression groups for histological analysis and Western assays after mice were sacrificed. STAT3 and YAP IHC staining was performed to determine the late effect of STAT3 plasmid transfection. The results show that STAT3 and nuclear YAP IHC staining increased in STAT3-overexpressing tissue compared to control tumor tissue (Figure 5C). WB showed that STAT3-overexpressing tumors had increased STAT3, p-STAT3, YAP, N-cadherin, and vimentin protein expression when compared with control tumors. STAT3-overexpressing tumors had decreased epithelial marker E-cadherin protein expression compared with control tumors (Figure 5D). Our results demonstrate that STAT3 overexpression increased STAT3 and nuclear YAP IHC staining and protein expression in H146 mouse xenograft tumor tissue. In addition, STAT3 overexpression promoted EMT in a mouse xenograft.

## 4. Discussion

Our study provides several lines of evidence suggesting that STAT3 interacts with YAP to promote the invasion and proliferation abilities of human SCLC cells. First, we showed that H209 cells had concurrently increased p-STAT3 and YAP protein expression compared with H146, H446 and H720 cells. Second, we found that forced overexpression of STAT3 in H146 and H446 cells increased YAP protein expression and mRNA expression of the YAP signaling downstream genes CTGF and CYR61. Third, we showed that the forced overexpression of STAT3 promoted EMT and the invasion ability of H146 and H446 cells. In addition, we demonstrated that the forced overexpression of STAT3 in H146 cells increased the tumor proliferation ability in a mouse xenograft model.

A previous study reported that the growth morphology is different between YAP-positive and YAP-negative SCLC cell lines [17]. In the same study, YAP knockdown in YAP-positive SCLC cells changed the adherent growth patterns to floating growth patterns with regard to morphology. Our study demonstrated that the YAP-positive SCLC cell line exhibited an adherent growth pattern, and the YAP-negative SCLC cell lines exhibited a floating growth pattern, which is compatible with the results of a previous study.

In contrast to the study conducted by Horie et al. [17], we showed that forced overexpression of STAT3 increased YAP expression and changed growth patterns from floating to adherent in YAP-negative SCLC cells. Previous studies reported that YAP protein expression in cancer cells is influenced by cell confluency, and the same level of cell growth confluency in cell lines is suggested in WB for YAP protein expression assessment [21,22]. The cell lines used in these previous studies were from epithelial cancer and malignant pleural mesothelioma and grew in an adherent pattern [21,22]. Different from these previous studies, the H146 and H446 cells used in our study grew in a floating pattern. Other previous studies also showed that floating-growth SCLC cell lines had extremely lower YAP protein expression than adherent-growth SCLC cell lines [17,23,24,25]. Taken together, this indicates that forced overexpression of STAT3 changes the morphology of SCLC cells through the activation of YAP signaling, which is not influenced by the cell confluency. Previous studies reported that SCLC cell lines with an adherent growth morphology had increased EMT markers and hallmark genes compared with SCLC cell lines with a floating growth morphology. The results of two studies indicated that adherent-growth SCLC cell lines had increased invasion and migration abilities compared with floating-growth SCLC cell lines [23,24,25]. The adherent growth transformation in STAT3-overexpressing H146 and H446 cells suggests that STAT3 overexpression increased migration and invasion abilities. In addition, the STAT3-overexpressing H146 xenograft tumors had increased EMT markers compared to the control H146 cells in our in vivo experiment.

Previous studies showed that positive YAP expression was associated with chemotherapy resistance in SCLC cell lines. In human tissue sample analysis, studies found that YAP1-negative patients were more chemosensitive than YAP1-positive patients [17,23,24,25]. In vitro experiments by Ito et al. also showed that YAP-negative SCLC cells were more sensitive to cisplatin treatment than YAP-positive SCLC cells. AJUBA is an oncoprotein reported to negatively regulate the Hippo kinase LATS1, and the inhibition of LATS1 by AJUBA induces YAP activation in cervical cancer and CRC [26,27,28]. The study conducted by Horie et al. also showed that AJUBA activates YAP expression by negatively regulating the Hippo kinase LATS1 in SCLC cells [17]. The alkylating DNA agent cisplatin and topoisomerase inhibitor etoposide are the chemotherapy regimens most frequently used for the treatment of SCLC. The concurrent activation of AJUBA and YAP protects cancer cells from DNA damage and promotes resistance to chemotherapy, as concluded by previous studies [26,27,28,29]. Other studies demonstrated that STAT3 was associated with chemoresistance in SCLC [10,11]. Peng et al. showed that inhibition of EPHA3 enhanced the cytotoxicity of chemotherapy to SCLC cells by suppressing STAT3 expression [10]. Another study showed that STAT3 expression was increased in SCLC cell lines with chemoresistance compared with their parental cell lines [11]. Taken together, the findings of our study may suggest that STAT3 promotes chemoresistance in SCLC cells through the activation of the YAP signaling pathway by suppressing the Hippo kinase LATS1.

The localization of YAP and phosphorylated YAP (pYAP) was often assayed to determine the activation of YAP in previous studies [30,31]. In the cytoplasm, the Hippo kinases degrade YAP at phosphorylated serine 127, and the pYAP(S127)-to-total YAP ratio has been used to determine the quantitation of nuclear YAP [30,31]. In this study, we did not assay the pYAP protein expression because the total YAP protein expression in the SCLC cell lines H146 and H446 was extremely low. Previous studies also showed that total YAP protein expression was extremely low in the same SCLC cell lines, and the pYAP protein expression was not assayed in these studies [17,23,24,25]. In our in vivo results, the nuclear YAP IHC staining did increase in STAT3-overexpressing tissue compared to control tumor tissue, and this result suggests that YAP translocated into the nucleus to activate downstream signaling. 

To our knowledge, we are the first to demonstrate that the interaction of STAT3 and YAP signaling increases the invasion and proliferation abilities of SCLC cells. CTGF and CYR61 are two well-known downstream genes regulated by YAP in human non-small cell lung cancer (NSCLC) cells [13,14,15], but the regulation of CTGF and CYR61 expression by YAP in SCLC cells has not been reported in previous studies. Increasing levels of the transcriptional downstream genes CTGF and CYR61 through the activation of YAP form autocrine loops with the ERBB and integrin pathways. In addition, the activating autocrine loops enforce mitogen-activated protein kinase (MAPK) and the AKT signaling pathways to promote EMT, invasion and proliferation in cancer cells [32,33]. We showed that the mRNA levels of the YAP downstream genes CTGF and CYR61 were significantly increased after STAT3 overexpression in H146 and H446 cells. This result suggests that STAT3 overexpression increased CTGF and CYR61 by activating YAP signaling in SCLC cells. Previous studies have shown that *E*-cadherin, *N*-cadherin, vimentin, MMP2, and MMP9 expression levels can indicate EMT and invasion in various cancers [34,35]. The changes in *E*-cadherin, *N*-cadherin, vimentin, MMP2, and MMP9 expression levels in our experimental results indicate that STAT3 overexpression promoted EMT in SCLC cells. The results of the Transwell assay in this study also indicate that STAT3 overexpression increased the invasion ability of SCLC cells. In addition, our mouse model indicated that overexpression of STAT3 promoted tumor proliferation by activating YAP signaling. Although STAT3 and YAP have been reported to promote EMT, invasion and proliferation in various cancers [36,37], previous studies investigating STAT3 in SCLC have mainly focused on drug resistance and not EMT, invasion or tumor progression. A previous study by Chaib et al. reported that the coactivation of STAT3 and YAP signaling increased resistance to epidermal growth factor (EGFR)-tyrosine kinase inhibitor (TKI) therapy in EGFR-mutated NSCLC [38]. In a cohort analysis of the same study, patients with concurrent increased STAT3 and YAP expression levels had a worse prognosis. Chaib et al. showed that dual inhibition of STAT3 and YAP synergized the cytotoxicity of gefitinib in their in vitro and in vivo experiments. Our experimental results show that the forced overexpression of STAT3 activates YAP and its downstream genes CTGF and CYR61, but inhibiting STAT3 by siRNA did not decrease YAP, CTGF and CYR61 mRNA expressions in the SCLC cell line H209 with concurrent STAT3 and YAP expression. The findings of our study suggest that STAT3 promotes EMT, invasion, and proliferation in SCLC cells through YAP and its downstream genes, but a single inhibition of STAT3 does not downregulate the YAP signaling pathway in SCLC cells with activation of YAP. The experimental findings of our study are compatible with those shown in the study of Chaib et al. [38], and STAT3 promotes EMT, invasion, and proliferation in SCLC cells by activating the YAP signaling pathway concurrently.

There is a limitation in this study because the SCLC samples used were all cell lines which were not the most representative to reproduce cancer features. Ideally, the use of SCLC patient-derived primary cell cultures may be better for understanding the real molecular biology and tumorigenesis in human SCLC studies. SCLC primary cultures are suggested to be closer to the tumor microenvironment than cell lines, and to exhibit crosstalk between malignant and normal cells as well [39]. A previous study successfully established SCLC patient-derived circulating tumor cell (CTC) spheroids, and they used this model to assess drug sensitivity in SCLC patients [40]. Another previous study established an SCLC orthotopic transplantation mouse model and showed that the SCLC orthotopic transplantation mouse model mimics the clinical metastatic tropism and oncogenic signaling of SCLC patients [41]. However, the establishment of patient-derived primary cell cultures and orthotopic transplantation mouse models takes a long time. The results of our experimental data can be verified again by using patient-derived primary cell cultures and even orthotopic transplantation mouse models in the future.

## 5. Conclusions

The results of our study suggest that the interaction between STAT3 and the YAP signaling pathway is involved in promoting EMT, invasion and proliferation in human SCLC. Future works investigating the interactions of STAT3 can be expanded by using patient-derived primary cell cultures, orthotopic transplantation mouse models and SCLC human tissue samples. In addition, future studies investigating the development of new therapies targeting the STAT3–YAP axis are warranted.

## Figures and Tables

**Figure 1 biomedicines-10-01704-f001:**
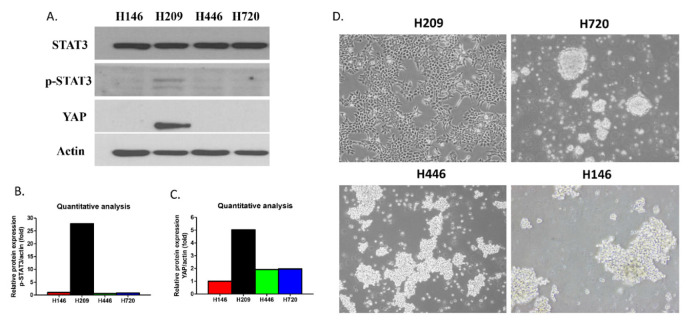
STAT3, p-STAT3 and YAP protein expression and morphology in human small cell lung cancer (SCLC) cell lines. (**A**) STAT3, p-STAT3 and YAP protein expression in SCLC cell lines. (**B**,**C**) Quantitative analysis of p-STAT3 and YAP protein expression in SCLC cell lines. (**D**) Morphology of SCLC cell lines.

**Figure 2 biomedicines-10-01704-f002:**
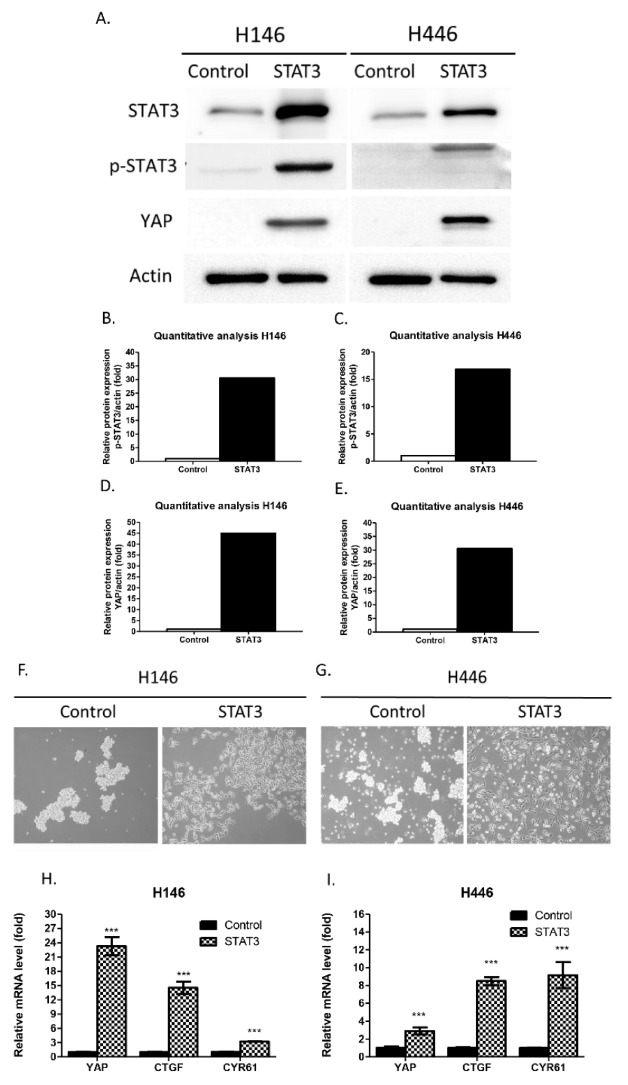
Forced overexpression of STAT3 increased YAP protein, YAP mRNA and YAP signaling downstream CTGF and CYR61 mRNA expression in H146 and H446 cells. (**A**–**E**) The STAT3, p-STAT3 and YAP protein expression levels increased in STAT3-overexpressing H146 and H446 cells compared with control H146 and H446 cells. (**F**,**G**) The floating cell morphology was changed to an adherent cell morphology after STAT3 overexpression in H146 and H446 cells. (**H**,**I**) YAP, CTGF and CYR61 mRNA expression significantly increased in STAT3-overexpressing H146 and H446 cells compared with control H146 and H446 cells. (Error bars indicate standard deviations; *** *p* ≤ 0.001.)

**Figure 3 biomedicines-10-01704-f003:**
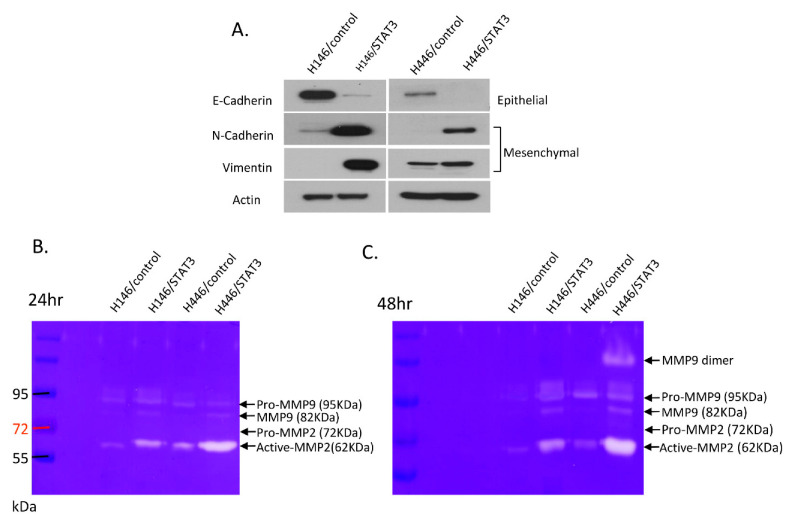
Epithelial–mesenchymal transition (EMT) marker expression changes after STAT3 overexpression in H146 and H446 cells. (**A**) E-cadherin protein expression decreased and N-cadherin and vimentin protein expression increased in STAT3-overexpressing H146 and H446 cells compared with control H146 and H446 cells. (**B**,**C**) STAT3-overexpressing H146 and H446 cells had increased matrix metalloproteinase-2 (MMP2) and matrix metalloproteinase-9 (MMP9) expression compared with control H146 and H446 cells.

**Figure 4 biomedicines-10-01704-f004:**
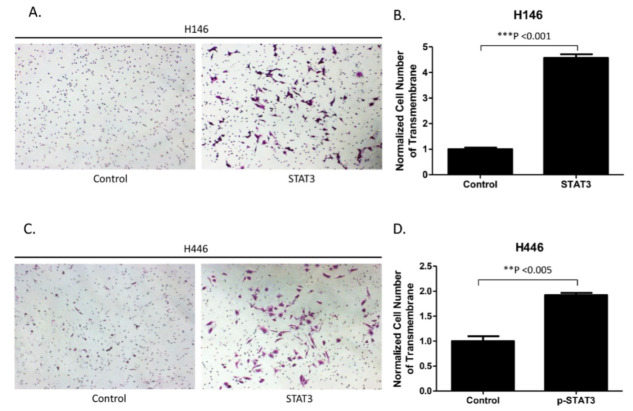
Analysis of the cell invasion ability of H146 and H446 cells after STAT3 overexpression. (**A**) Increase in the cell invasion ability of H146 cells after STAT3 overexpression. (**B**) Quantitative analysis of Transwell invasion assay results, which indicate that STAT3 overexpression increased the invasion ability of H146 cells. (**C**) Increase in the cell invasion ability of H446 cells after STAT3 overexpression. (**D**) Quantitative analysis of Transwell invasion assay results, indicating that STAT3 overexpression increased the invasion ability of H446 cells. (Error bars indicate standard deviations; ** *p* ≤ 0.01, *** *p* ≤ 0.001.).

**Figure 5 biomedicines-10-01704-f005:**
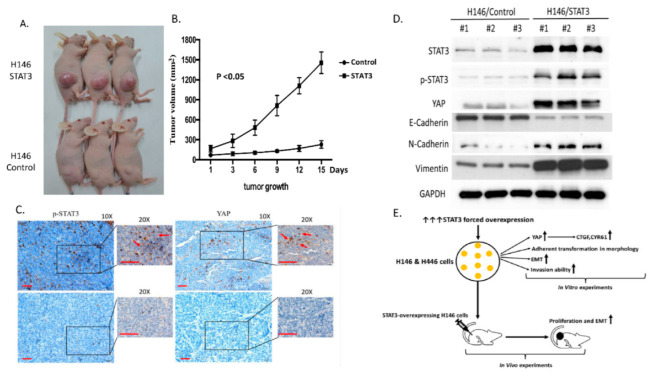
STAT3 overexpression increased the tumor proliferation ability of H146 cells in a xenograft mouse model. (**A**,**B**) STAT3-overexpressing H146 cell tumors grew significantly more rapidly than control H146 cell tumors (*p* < 0.05). (**C**) Histological analysis of mouse xenograft tissue showed that p-STAT3 and YAP immunohistochemistry (IHC) staining increased in STAT3-overexpressing H146 cell tumors compared with control H146 cell tumors. Scale bar = 100 µm. (**D**) WB of STAT3, p-STAT3, YAP, E-cadherin, N-cadherin and vimentin protein expressions in STAT3-overexpressing and control tumors. (**E**) Experimental framework scheme of this study.

## Data Availability

Data is contained within the article and Appendix A.

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
