# Peer review of "Forced Overexpression of Signal Transducer and Activator of Transcription 3 (STAT3) Activates Yes-Associated Protein (YAP) Expression and Increases the Invasion and Proliferation Abilities of Small Cell Lung Cancer (SCLC) Cells"

_biomedicines, 2022, doi:10.3390/biomedicines10071704_

Round 1

Reviewer 1 Report

The authors of the present work studied the interactions between STAT3 and YAP on human SCLC cell lines. The results show the activation of EMT processes and an increased migration capability of lung cancer cells overexpressing STAT3. Moreover, the forced expression of STAT3 induced a high proliferation of the SCLC cells in the xenograft mouse model.   

The manuscript looks like well written and organized. The authors have presented an interesting topic in the field of lung cancer. The paper should be considered after major revisions.

1.       The paragraph 3.4 described the results of the in vitro migration experiments. The behavior of the two cell lines H146 and H446 looks the same. Indeed, the graphs of Figure 4 (B and D) are almost identical, and the title of the graph D reports H146 instead of H446. Are there any mistakes? The authors should briefly describe the results of these experiments in the text.

2.       The authors performed transwell assays to demonstrate the increased migration capability induced by STAT3 overexpression. It could be not enough to demonstrate this altered cell behavior. The authors should use another in vitro or in vivo model to confirm these data.

3.       Cell lines represent the kind of sample major employed in cancer research but not the most representative to reproduce cancer features. The authors should discuss about the importance of the use of patient-derived primary cultures to better understand the biology and malignant processes of SCLC. The reference entitled: "Management and Potentialities of Primary Cancer Cultures in Preclinical and Translational Studies” doi: 10.1186/s12967-017-1328-z, or other with similar considerations should be added to the manuscript.

Author Response

Response to Reviewer 1 Comments

Point 1: The paragraph 3.4 described the results of the in vitro migration experiments. The behavior of the two cell lines H146 and H446 looks the same. Indeed, the graphs of Figure 4 (B and D) are almost identical, and the title of the graph D reports H146 instead of H446. Are there any mistakes? The authors should briefly describe the results of these experiments in the text.

Response 1: We had corrected the mistakes in figure 4B and 4D, and added a description of the results in revised manuscript as suggested.

Transwell assays showed that the number of cells invading the lower side of membranes obviously increased in STAT3-overexpressing H146 and H446 cells compared with control H146 and H446 cells by crystal violet staining (Figure 4A,C). In quantitative analysis of the Transwell invasion assay, the number of cells invading the lower side of membranes significantly increased in STAT3-overexpressing H146 and H446 cells (P<0.05) (Figure 4B,D).

Point 2: The authors performed transwell assays to demonstrate the increased migration capability induced by STAT3 overexpression. It could be not enough to demonstrate this altered cell behavior. The authors should use another in vitro or in vivo model to confirm these data.

Response 2: Regarding the concern about “It could be not enough to demonstrate this altered cell behavior and needs another in vitro or in vivo model to confirm these data.”, the experimental data of figure 2F, 2G and 5D in this manuscript can also support that STAT3 overexpression increased migration and invasion abilities.

   Previous studies reported that SCLC cell lines with an adherent growth morphology had increased EMT markers and hallmark genes compared with SCLC cell lines with a floating growth morphology. The results of two studies indicated that adherent-growth SCLC cell lines had increased invasion and migration abilities compared with floating-growth SCLC cell lines [1,2]. The adherent growth transformation in STAT3-overexpressing H146 and H446 cells suggests that STAT3 overexpression increased migration and invasion abilities. In addition, the STAT3-overexpressing H146 xenograft tumors had increased EMT markers compared to the control H146 cells in our in vivo experiment.

      We had added a paragraph to discuss this point in revised manuscript.

Point 3: Cell lines represent the kind of sample major employed in cancer research but not the most representative to reproduce cancer features. The authors should discuss about the importance of the use of patient-derived primary cultures to better understand the biology and malignant processes of SCLC. The reference entitled: "Management and Potentialities of Primary Cancer Cultures in Preclinical and Translational Studies” doi: 10.1186/s12967-017-1328-z, or other with similar considerations should be added to the manuscript.

Response 3: We had added a paragraph to discuss this point as suggested, and cite some references.

There is a limitation in this study because the SCLC samples used were all cell lines which were not the most representative to reproduce cancer features. Ideally, the use of SCLC patient-derived primary cell cultures may be better for understanding the real molecular biology and tumorigenesis in human SCLC studies. SCLC primary cultures are suggested to be closer to the tumor microenvironment than cell lines, and to exhibit crosstalk between malignant and normal cells as well [3]. A previous study successfully established SCLC patient-derived circulating tumor cell (CTC) spheroids, and they used this model to assess drug sensitivity in SCLC patients [4]. Another previous study established an SCLC orthotopic transplantation mouse model and showed that the SCLC orthotopic transplantation mouse model mimics the clinical metastatic tropism and oncogenic signaling of SCLC patients [5]. However, the establishment of patient-derived primary cell cultures and orthotopic transplantation mouse models takes a long time. The results of our experimental data can be verified again by using patient-derived primary cell cultures and even orthotopic transplantation mouse models in the future.

Reference

  1. Krohn, A.; Ahrens, T.; Yalcin, A.; Plönes, T.; Wehrle, J.; Taromi, S.; Wollner, S.; Follo, M.; Brabletz, T.; Mani, S.A.; et al. Tumor cell heterogeneity in Small Cell Lung Cancer (SCLC): phenotypical and functional differences associated with Epithelial-Mesenchymal Transition (EMT) and DNA methylation changes. PLoS One. 2014 Jun 24;9(6):e100249.

  1. Zhang, W.; Girard, L.; Zhang, Y.A.; Haruki, T.; Papari-Zareei, M.; Stastny, V.; Ghayee, H.K.; Pacak, K.; Oliver, T.G.; Minna, J.D.; et al. Small cell lung cancer tumors and preclinical models display heterogeneity of neuroendocrine phenotypes. Transl Lung Cancer Res. 2018 Feb;7(1):32-49.

  1. Miserocchi, G.; Mercatali, L.; Liverani, C.; De Vita, A.; Spadazzi, C.; Pieri, F.; Bongiovanni, A.; Recine, F.; Amadori, D.; Ibrahim, T. Management and potentialities of primary cancer cultures in preclinical and translational studies. J Transl Med. 2017 Nov 7;15(1):229.

  1. Lee, H.-L.; Chiou, J.-F.; Wang, P.-Y.; Lu, L.-S.; Shen, C.-N.; Hsu, H.-L.; Burnouf, T.; Ting, L.-L.; Chou, P.-C.; Chung, C.-L.; Lee, K.-L.; Shiah, H.-S.; Liu, Y.-L.; Chen, Y.-J. Ex Vivo Expansion and Drug Sensitivity Profiling of Circulating Tumor Cells from Patients with Small Cell Lung Cancer. Cancers 2020, 12, 3394.

  1. Sakamoto, S.; Inoue, H.; Ohba, S.; Kohda, Y.; Usami,I.; Masuda, T.; Kawada, M.; Nomoto, A. New metastatic model of human small-cell lung cancer by orthotopic transplantation in mice. Cancer Sci. 2015 Apr;106(4):367-74.

Reviewer 2 Report

This paper provides data regarding a possible interaction between YAP and STAT3. It has already been shown by different groups that STAT3 cooperates with YAP (transcriptional co-activator) to regulate the expression of a number of target genes. Herein the authors describe a potential reverse mechanism by which activated STAT3 regulates YAP expression.

Comments: Given the fact that, under canonical conditions, YAP is involved in cell proliferation, stem cell regulation and organ development and regeneration, it is expected that Hippo pathway is implicated in tumorigenesis. Further establishing the functional relevance of elevated YAP levels in cancer, nuclear YAP is more common in neoplastic tissues than cytoplasmic YAP which is
found in 85% of normal tissues. In line with these observations, a growing amount of evidence support that YAP can exert both protumorigenic and tumor suppressor effects depending on the cellular localization and YAPs interacting partners.Previous studies have reported that there are Hippo-dependent (through YAP phosphorylation) and Hippo-independent mechanisms that regulate YAP activity. The authors have suggested that upon STAT3 overexpression, there is a transcriptional increase in YAP expression. However, the mechanism by which STAT3 overexpression leads to an increase in YAP expression is not shown. Based on these it would be important to show YAP localization in your cell lines and also show pYAP expression as well. Moreover the authors should show the expression of the other core kinases of the HIPPO pathway (LATS, MST, etc). Furthermore, as YAPs expression is influenced by cell confluency, the authors should grow all cell lines to the same level of confluency and then measure with WB YAP expression (usually 70% confluency). Based on this, the results shown in pic 1, can be explained by different confluency and not through STAT3 expression. Moreover, they should use a STAT3 inhibitor and show the effect on YAP expression as well. Finally there are reports showing that CTGF and CYR61 are also regulated by STAT3. In order to state that these are downstream of YAP more studies should be done (si RNA YAP and pharmacologic inhibition of YAP as well as IP-immunoprecipitation studies).

Author Response

Response to Reviewer 2 Comments

Point 1: The authors have suggested that upon STAT3 overexpression, there is a transcriptional increase in YAP expression. However, the mechanism by which STAT3 overexpression leads to an increase in YAP expression is not shown.

Response 1: In response to the concern about “the mechanism by which STAT3 overexpression leads to an increase in YAP expression is not shown.”, we had assayed the Hippo kinase LATS1 protein change in STAT3-overexpressing H146 and H446 cells by western blotting.

To verify the mechanism of how STAT3 increased YAP expression, the Hippo kinase LATS1 was assayed by WB. WB showed that LATS1 protein expression decreased in STAT3-overexpressing H146 and H446 cells compared with control H146 and H446 cells (Supplementary Figure S1A-–C). 

The results indicated that, in H146 and H446 cells, forced overexpression of STAT3 activates YAP protein by suppressing Hippo kinase LATS1. We added a paragraph to discuss about the Hippo kinase in introduction section to address the reviewer’s comment.

YAP is negatively regulated by Hippo kinases such as Merlin, large tumor suppressor homolog 1 (LATS1), LATS2 and mammalian sterile 20-like kinase 1 (MST1). In the usual manner, the Hippo kinases sequester and degrade YAP in the cytoplasm. Conversely, when the Hippo kinase is deregulated, the increasing YAP translocates from the cytoplasm into the nucleus and binds to transcriptional enhancer factors (TEAD). The binding of YAP and TEAD forms a complex enhancing the downstream gene transcription to promote cell proliferation, EMT, invasion and drug resistance

      The results were shown in supplementary figure S1A-S1C.

Point 2: Based on these it would be important to show YAP localization in your cell lines and also show pYAP expression as well.

Response 2: In response to the concern about “YAP localization in cell lines and pYAP expression”, we had shown that nuclear YAP IHC staining increased in STAT3-overexpressing tissue compared to control tumor tissue in Figure 5C. We also added a paragraph in discussion section to explain that why pYAP was not shown in this manuscript.

    The localization of YAP and phosphorylated YAP (pYAP) was often assayed to determine the activation of YAP in previous studies [30,31]. In the cytoplasm, the Hippo kinases degrade YAP at phosphorylated serine 127, and the pYAP(S127)-to-total YAP ratio has been used to determine the quantitation of nuclear YAP [1,2]. In this study, we did not assay the pYAP protein expression because the total YAP protein expression in the SCLC cell lines H146 and H446 was extremely low. Previous studies also showed that total YAP protein expression was extremely low in the same SCLC cell lines, and the pYAP protein expression was not assayed in these studies [3-6]. In our in vivo results, the nuclear YAP IHC staining did increase in STAT3-overexpressing tissue compared to control tumor tissue, and this result suggests that YAP translocated into the nucleus to activate downstream signaling.

Point 3: Moreover the authors should show the expression of the other core kinases of the HIPPO pathway (LATS, MST, etc).

Response 3: In response to the comment about “the expression of the other core kinases of the HIPPO pathway”, we had added the result in supplementary figure S1A-1C as suggested.

To verify the mechanism of how STAT3 increased YAP expression, the Hippo kinase LATS1 was assayed by WB. WB showed that LATS1 protein expression decreased in STAT3-overexpressing H146 and H446 cells compared with control H146 and H446 cells (Supplementary Figure S1A-–C).

Point 4: Furthermore, as YAPs expression is influenced by cell confluency, the authors should grow all cell lines to the same level of confluency and then measure with WB YAP expression (usually 70% confluency). Based on this, the results shown in pic 1, can be explained by different confluency and not through STAT3 expression.

Response 4: Regarding to the concern about “the results shown in pic 1, can be explained by different confluency and not through STAT3 expression”, we had added a paragraph to explain this point in discussion section of revised manuscript.

In contrast to the study conducted by Horie et al. [3], we showed that forced overexpression of STAT3 increased YAP expression and changed growth patterns from floating to adherent in YAP-negative SCLC cells. Previous studies reported that YAP protein expression in cancer cells is influenced by cell confluency, and the same level of cell growth confluency in cell lines is suggested in WB for YAP protein expression assessment [7,8]. The cell lines used in these previous studies were from epithelial cancer and malignant pleural mesothelioma and grew in an adherent pattern [7,8]. Different from these previous studies, the H146 and H446 cells used in our study grew in a floating pattern. Other previous studies also showed that floating-growth SCLC cell lines had extremely lower YAP protein expression than adherent-growth SCLC cell lines [3-6]. Taken together, this indicates that forced overexpression of STAT3 changes the morphology of SCLC cells through the activation of YAP signaling, which is not influenced by the cell confluency.

Point 5: Moreover, they should use a STAT3 inhibitor and show the effect on YAP expression as well.

Response 5: We had added the results of YAP, CTGF and CYR61 mRNA expression after STAT3 knock down by siRNA in revised manuscript as suggested.

The changes in YAP mRNA and YAP signaling downstream genes CTGF and CYR61 in H209 cells after STAT3 knockdown by siRNAs were also assayed by qRT–PCR. STAT3 knockdown by siRNAs did not decrease the mRNA level of YAP, CTGF and CYR61 expressions in H209 cells (Supplementary Figure S1D–F).

Point 6: Finally there are reports showing that CTGF and CYR61 are also regulated by STAT3. In order to state that these are downstream of YAP more studies should be done (si RNA YAP and pharmacologic inhibition of YAP as well as IP-immunoprecipitation studies).

Response 6: In response to the comment about that “more studies should be done (si RNA YAP and pharmacologic inhibition of YAP as well as IP-immunoprecipitation studies)”, this study mainly focused on the effect of overexpression of STAT3, not on the inhibition of STAT3 or YAP.

We had added a paragraph in revised manuscript to address this comment.

A previous study by Chaib et al. reported that the coactivation of STAT3 and YAP signaling increased resistance to epidermal growth factor (EGFR)-tyrosine kinase inhibitor (TKI) therapy in EGFR-mutated NSCLC [38]. In a cohort analysis of the same study, patients with concurrent increased STAT3 and YAP expression levels had a worse prognosis. Chaib et al. showed that dual inhibition of STAT3 and YAP synergized the cytotoxicity of gefitinib in their in vitro and in vivo experiments. Our experimental results show that forced overexpression of STAT3 activates YAP and its downstream genes CTGF and CYR61, but inhibiting STAT3 by siRNA did not decrease YAP, CTGF and CYR61 mRNA expressions in the SCLC cell line H209 with concurrent STAT3 and YAP expression. The findings of our study suggest that STAT3 promotes EMT, invasion and proliferation in SCLC cells through YAP and its downstream genes, but a single inhibition of STAT3 does not downregulate the YAP signaling pathway in SCLC cells with activation of YAP. The experimental findings of our study are compatible with those shown in the study of Chaib et al. [9], and STAT3 promotes EMT, invasion and proliferation in SCLC cells by activating the YAP signaling pathway concurrently.

Reference:

1.Hsu, P.-C.; Yang, C.-T.; Jablons, D.M.; You, L. The Crosstalk between Src and Hippo/YAP Signaling Pathways in Non-Small Cell Lung Cancer (NSCLC). Cancers 2020, 12, 1361.

2.Kalita-de Croft, P.; Lim, M.; Chittoory, H.; de Luca, X.M.; Kutasovic, J.R.; Day, B.W.; Al-Ejeh, F.; Simpson, P.T.; McCart Reed, A.E.; Lakhani, S.R.; Saunus, J.M. Clinicopathologic significance of nuclear HER4 and phospho-YAP(S127) in human breast cancers and matching brain metastases. Ther Adv Med Oncol. 2020 Jul 31;12:1758835920946259.

3.Horie, M.; Saito, A.; Ohshima, M.; Suzuki, H.I.; Nagase, T. YAP and TAZ modulate cell phenotype in a subset of small cell lung cancer. Cancer Sci. 2016 Dec;107(12):1755-1766.

4.Krohn, A.; Ahrens, T.; Yalcin, A.; Plönes, T.; Wehrle, J.; Taromi, S.; Wollner, S.; Follo, M.; Brabletz, T.; Mani, S.A.; et al. Tumor cell heterogeneity in Small Cell Lung Cancer (SCLC): phenotypical and functional differences associated with Epitheli-al-Mesenchymal Transition (EMT) and DNA methylation changes. PLoS One. 2014 Jun 24;9(6):e100249.

5.Zhang, W.; Girard, L.; Zhang, Y.A.; Haruki, T.; Papari-Zareei, M.; Stastny, V.; Ghayee, H.K.; Pacak, K.; Oliver, T.G.; Minna, J.D.; et al. Small cell lung cancer tumors and preclinical models display heterogeneity of neuroendocrine phenotypes. Transl Lung Cancer Res. 2018 Feb;7(1):32-49.

6.Ito, T.; Matsubara, D.; Tanaka, I.; Makiya, K.; Tanei, Z.I.; Kumagai, Y.; Shiu, S.J.; Nakaoka, H.J.; Ishikawa, S.; Isagawa, T.; et al. Loss of YAP1 defines neuroendocrine differentiation of lung tumors. Cancer Sci. 2016 Oct;107(10):1527-1538.

  1. Wu, J.; Minikes, A.M.; Gao, M.; Bian, H.; Li, Y.; Stockwell, B.R.; Chen, Z.N.; Jiang, X. Intercellular interaction dictates cancer cell ferroptosis via NF2-YAP signalling. Nature. 2019 Aug;572(7769):402-406.
  2. Sun, T.; Chi, J.T. Regulation of ferroptosis in cancer cells by YAP/TAZ and Hippo pathways: The therapeutic implications. Genes Dis. 2020 May 18;8(3):241-249.
  3. Chaib, I.; Karachaliou, N.; Pilotto, S.; Codony Servat, J.; Cai, X.; Li, X.; Drozdowskyj, A.; Servat, C.C.; Yang, J.; Hu, C.; et al. Co-activation of STAT3 and YES-Associated Protein 1 (YAP1) Pathway in EGFR-Mutant NSCLC. J. Natl. Cancer Inst. 2017, 109, djx014.

Reviewer 3 Report

The article entitled "Forced Overexpression of Signal Transducer and Activator of Transcription 3 (STAT3) Activates Yes-Associated Protein (YAP) Expression and Increases the Invasion and Proliferation Abilities of Small Cell Lung Cancer (SCLC) Cells". The authors have sought to investigate the interaction between STAT3 and YAP signaling pathway in human SCLC. For which authors have done some series of in vivo and in vitro experiments. The study results suggested that, Overexpression of STAT3 elevates the SCLC Epithelial-mesenchymal transition (EMT), invasion and proliferation via the activation of the YAP signaling pathways.

The manuscript comprises all the necessary elements of scientific novelty. The experimental designing and execution of the study were appreciable. I recommend this article for publication after incorporating changes given in below.

The words In vivo and in vitro should be in italics. Check and rectify it throughout the manuscript.

Line 35- 10%-15% should be 10-15%

In introduction section lines 67-68 needs to explain more in three to four lines. It some what covers your experimental part too.

Connecting/ conjunctions are needed between STAT3 and YAP para of introduction section.

Authors must concentrate on the formatting, and use of symbols, etc.,

In materials and method section: authors should explain why each item of methodology was done.

Framework figure is required. It will be useful to the readers for better understanding of the studied issue.

Line 120: quantitative real-time polymerase chain reaction (RT–PCR) à  it is a (qRT-PCR). Real time PCR (qRT-PCR/qPCR) and RT-PCR is different check and rectify it in throughout the manuscript.

Figure 1 B and C should be improved.

Line 218: Remove epithelial–mesenchymal transition à EMT is enough. First mention only needs full acronym. Follow the same in throughout the manuscript.

In figure 3: Why there is no arrow mark for Pro-MMP2 (72kDa)? Check it.

Discussion section looks shallow. It should be discussed more.

Conclusion section needs to be improvised and add few lines about future perspectives and hypothesize the current study. It will be useful to the readers community to design and understand the importance of studied issue.

Author Response

Response to Reviewer 3 Comments

Point 1: The words In vivo and in vitro should be in italics. Check and rectify it throughout the manuscript.

Response 1: We had corrected it in revised manuscript as suggested.

Point 2: Line 35- 10%-15% should be 10-15%

Response 2: We corrected this error as suggested in revised manuscript.

Point 3: In introduction section lines 67-68 needs to explain more in three to four lines. It some what covers your experimental part too.

Response 3: We had added a paragraph to expand this discussion as suggested.

Previous studies reported that STAT3 interacted with YAP signaling in head and neck squamous cell carcinoma (HNSCC) and colorectal cancer (CRC). Li et al. showed that STAT3 activated YAP to promote tumor progression through suppressing the Hippo kinase LATS1 [1]. Another study conducted by Shen et al. concluded that the interaction of the STAT3–YAP axis plays a crucial role in angiogenesis in CRC cells [2]. Whether STAT3 crosstalks with the YAP signaling pathway in SCLC is unclear. In this study, we sought to explore the interaction between STAT3 and YAP, and the impacts of overexpression of STAT3 on YAP signaling downstream genes, EMT, invasion and proliferation in human SCLC cells.

Point 4: Connecting/ conjunctions are needed between STAT3 and YAP para of introduction section.

Response 4: We had added a paragraph to connect STAT3 and YAP in revised manuscript as suggested.

Previous studies reported that STAT3 interacted with YAP signaling in head and neck squamous cell carcinoma (HNSCC) and colorectal cancer (CRC). Li et al. showed that STAT3 activated YAP to promote tumor progression through suppressing the Hippo kinase LATS1 [1]. Another study conducted by Shen et al. concluded that the interaction of the STAT3–YAP axis plays a crucial role in angiogenesis in CRC cells [2]. Whether STAT3 crosstalks with the YAP signaling pathway in SCLC is unclear. In this study, we sought to explore the interaction between STAT3 and YAP, and the impacts of overexpression of STAT3 on YAP signaling downstream genes, EMT, invasion and proliferation in human SCLC cells.

Point 5: Authors must concentrate on the formatting, and use of symbols, etc.,

Response 5: We had done it as suggested

Point 6: In materials and method section: authors should explain why each item of methodology was done.

Response 6: We had added the explanations of why each item of methodology was done in revised manuscript as suggested.

Point 7: Framework figure is required. It will be useful to the readers for better understanding of the studied issue.

Response 7: We added a framework figure as figure 5E in revised manuscript as suggested.

Point 8: Line 120: quantitative real-time polymerase chain reaction (RT–PCR) à it is a (qRT-PCR). Real time PCR (qRT-PCR/qPCR) and RT-PCR is different check and rectify it in throughout the manuscript.

Response 8: We had corrected it in revised manuscript as suggested.

Point 9: Figure 1 B and C should be improved.

Response 9: We had replaced a new figure with better quality in revised manuscript as suggested.

Point 10: Line 218: Remove epithelial–mesenchymal transition à EMT is enough. First mention only needs full acronym. Follow the same in throughout the manuscript.

Response 10: We corrected it as suggested in revised manuscript.

Point 11: In figure 3: Why there is no arrow mark for Pro-MMP2 (72kDa)? Check it.

Response 11: We added the arrow mark for Pro-MMP2 (72kDa) in revised figure 3.

Point 12: Discussion section looks shallow. It should be discussed more.

Response 12: We had expanded the discussion and cited more references in revised manuscript as suggested.

Point 13: Conclusion section needs to be improvised and add few lines about future perspectives and hypothesize the current study. It will be useful to the readers community to design and understand the importance of studied issue.

Response 13: We had expanded the conclusion section in revised manuscript as suggested.

Reference

  1. Li, J.; Shi, C.; Zhou, R.; Han, Y.; Xu, S.; Ma, H.; Zhang, Z. The crosstalk between AXL and YAP promotes tumor progression through STAT3 activation in head and neck squamous cell carcinoma. Cancer Sci. 2020 Sep;111(9):3222-3235.

  1. Shen, Y.; Wang, X.; Liu, Y.; Singhal, M.; GürkaÅŸlar, C.; Valls, A.F.; Lei, Y.; Hu, W.; Schermann, G.; Adler, H.; et al. STAT3-YAP/TAZ signaling in endothelial cells promotes tumor angiogenesis. Sci Signal. 2021 Dec 7;14(712):eabj8393.

Reviewer 4 Report

This is an interesting study that  assessed the effect of overexpression of  STAT3) on YAP expression and cell invasion.

I just have a few questions for the authors:

1. Why were the specific cell lines used in this study selected as I could find no details about the cells or justification for their use in the study?

2. In lines 289/290 the authors describe the change from `floating to coherent cells,` what is the relevance of this for cancer growth/invasion?

3. How does the activation of the YAP signalling pathway (lines 308/9) promote chemoresistance?

4. What is the significance of the inc expression of CTGF/CYR61 reported in the study (line 307)?

5. What is the actual evidence that the` interaction between STAT3 and the YAP 331 signalling pathways was partly involved in cancer progression ( line331)` as it wasn`t clear from the results presented in the paper?

that 21 STAT3 overexpression promoted (Epithelial-mesenchymal transition) EMT

Author Response

Response to Reviewer 4 Comments

Point 1: Why were the specific cell lines used in this study selected as I could find no details about the cells or justification for their use in the study?

Response 1: Regarding the concern about “why were the specific cell lines used in this study selected”, we make the explanation in method section.

This study was performed using SCLC cell lines mainly. Most SCLC cell lines appear with a floating growth pattern (H146, H446 and H720), and H209 is a rare SCLC cell line with an adherent growth pattern. We selected these cell lines to verify whether there is different STAT3 and YAP expressions in SCLC with different growth patterns.

Point 2: In lines 289/290 the authors describe the change from `floating to coherent cells,` what is the relevance of this for cancer growth/invasion?

Response 2: Regarding the concern about “the change from `floating to coherent cells,` what is the relevance of this for cancer growth/invasion”, the adherent growth transformation in SCLC cell lines suggests increased migration and invasion abilities.

Previous studies reported that SCLC cell lines with an adherent growth morphology had increased EMT markers and hallmark genes compared with SCLC cell lines with a floating growth morphology. The results of two studies indicated that adherent-growth SCLC cell lines had increased invasion and migration abilities compared with floating-growth SCLC cell lines [1-3]. The adherent growth transformation in STAT3-overexpressing H146 and H446 cells suggests that STAT3 overexpression increased migration and invasion abilities. In addition, the STAT3-overexpressing H146 xenograft tumors had increased EMT markers compared to the control H146 cells in our in vivo experiment.

      We had added a paragraph to discuss this point in revised manuscript.

Point 3: How does the activation of the YAP signalling pathway (lines 308/9) promote chemoresistance?

Response 3: We added a paragraph to explain “how does the activation of the YAP signalling pathway promote chemoresistance” in revised manuscript as suggested.

AJUBA is an oncoprotein reported to negatively regulate the Hippo kinase LATS1, and the inhibition of LATS1 by AJUBA induces YAP activation in cervical cancer and CRC [3-5]. The study conducted by Horie et al. also showed that AJUBA activates YAP expression by negatively regulating the Hippo kinase LATS1 in SCLC cells [3]. The alkylating DNA agent cisplatin and topoisomerase inhibitor etoposide are the chemotherapy regimens most frequently used for the treatment of SCLC. The concurrent activation of AJUBA and YAP protects cancer cells from DNA damage and promotes resistance to chemotherapy, as concluded by previous studies

Point 4: What is the significance of the inc expression of CTGF/CYR61 reported in the study (line 307)?

Response 4: In response to the comment about “what is the significance of the inc expression of CTGF/CYR61 reported in the study”, we had discussed this point in revised manuscript.

Increasing levels of the transcriptional downstream genes CTGF and CYR61 through the activation of YAP form autocrine loops with the ERBB and integrin pathways. In addition, the activating autocrine loops enforce mitogen-activated protein kinase (MAPK) and the AKT signaling pathways to promote EMT, invasion and proliferation in cancer cells [5,6]. We showed that the mRNA levels of the YAP downstream genes CTGF and CYR61 were significantly increased after STAT3 overexpression in H146 and H446 cells. This result suggests that STAT3 overexpression increased CTGF and CYR61 by activating YAP signaling in SCLC cells.

Point 5: What is the actual evidence that the` interaction between STAT3 and the YAP 331 signalling pathways was partly involved in cancer progression (line331)` as it wasn`t clear from the results presented in the paper

Response 5: In response to the concern about “what is the actual evidence that the` interaction between STAT3 and the YAP 331 signalling pathways was partly involved in cancer progression (line331)` as it wasn`t clear from the results presented in the paper”, we had re-written this sentence to avoid confusing in revised manuscript.

The results of our study suggest that the interaction between STAT3 and the YAP signaling pathway is involved in promoting EMT, invasion and proliferation in human SCLC. Future works investigating the interactions of STAT3 can be expanded by using patient-derived primary cell cultures, orthotopic transplantation mouse models and SCLC human tissue samples. In addition, future studies investigating the development of new therapies targeting the STAT3–YAP axis are warranted.

Reference

  1. Krohn, A.; Ahrens, T.; Yalcin, A.; Plönes, T.; Wehrle, J.; Taromi, S.; Wollner, S.; Follo, M.; Brabletz, T.; Mani, S.A.; et al. Tumor cell heterogeneity in Small Cell Lung Cancer (SCLC): phenotypical and functional differences associated with Epithelial-Mesenchymal Transition (EMT) and DNA methylation changes. PLoS One. 2014 Jun 24;9(6):e100249.

  1. Zhang, W.; Girard, L.; Zhang, Y.A.; Haruki, T.; Papari-Zareei, M.; Stastny, V.; Ghayee, H.K.; Pacak, K.; Oliver, T.G.; Minna, J.D.; et al. Small cell lung cancer tumors and preclinical models display heterogeneity of neuroendocrine phenotypes. Transl Lung Cancer Res. 2018 Feb;7(1):32-49.

3.Horie, M.; Saito, A.; Ohshima, M.; Suzuki, H.I.; Nagase, T. YAP and TAZ modulate cell phenotype in a subset of small cell lung cancer. Cancer Sci. 2016 Dec;107(12):1755-1766.

4.Nguyen, C.D.K.; Yi, C. YAP/TAZ Signaling and Resistance to Cancer Therapy. Trends Cancer. 2019 May;5(5):283-296.

5.Wu, Q.; Guo, J.; Liu, Y.; Zheng, Q.; Li, X.; Wu, C.; Fang, D.; Chen, X.; Ma, L.; Xu, P.; et al. YAP drives fate conversion and chemoresistance of small cell lung cancer. Sci Adv. 2021 Oct;7(40):eabg1850.

  1. Zhu, X.; Zhong, J.; Zhao, Z.; Sheng, J.; Wang, J.; Liu, J.; Cui, K.; Chang, J.; Zhao, H.; Wong, S. Epithelial derived CTGF promotes breast tumor progression via inducing EMT and collagen I fibers deposition. Oncotarget. 2015 Sep 22;6(28):25320-38.

6.Shi, W.; Zhang, C.; Chen, Z.; Chen, H.; Liu, L.; Meng, Z. Cyr61-positive cancer stem-like cells enhances distal metastases of pancreatic cancer. Oncotarget. 2016 Nov 8;7(45):73160-73170.

Round 2

Reviewer 1 Report

The manuscript is now acceptable for the pubblication

This manuscript is a resubmission of an earlier submission. The following is a list of the peer review reports and author responses from that submission.